# Even Faster Hyperbolic Random Forests: A Beltrami-Klein Wrapper Approach

## Abstract

Decision trees and models that use them as primitives are workhorses of machine learning in Euclidean spaces. Recent work has further extended these models to the Lorentz model of hyperbolic space by replacing axis-parallel hyperplanes with homogeneous hyperplanes when partitioning the input space. In this paper, we show how the HYPERDT algorithm can be elegantly reexpressed in the Beltrami-Klein model of hyperbolic spaces. This preserves the thresholding operation used in Euclidean decision trees, enabling us to further rewrite HYPERDT as simple pre– and post-processing steps that form a wrapper around existing tree-based models designed for Euclidean spaces. The wrapper approach unlocks many optimizations already available in Euclidean space models, improving flexibility, speed, and accuracy while offering a simpler, more maintainable, and extensible codebase.

## 1 Introduction

All machine learning classifiers implicitly encode geometric assumptions about their feature space. Among these, standard decision trees and classifiers built on them stand out as being remarkably indifferent to geometry: Decision tree learning is ultimately a discrete optimization over label statistics, one feature at a time. The metric structure of the multi-feature space only matters when evaluating its final partition.

In contrast, all linear and neural models relying on matrix multiplication implicitly work with signed distances to a decision-boundary hyperplane; probabilistic methods based on Gaussian densities such as Naive Bayes have probability densities that depend on distance to the centroid; not to mention explicitly distance-based methods such as support vector machines (SVMs) and $k$-nearest neighbors. Yet decision trees rely only on label statistics for candidate partitions which depend only on the ordering of features along each dimension: a property we exploit to reformalize their behavior in hyperbolic space.

There are two existing approaches in the literature to extending decision trees to hyperbolic spaces. Doorenbos et al. (2023) proposes HORORF, an ensemble of horospherical support vector machines (Fan et al., 2023), equating between horospheres in hyperbolic and hyperplanes in Euclidean space, as well as between SVMs and linear splits at a decision tree node. In contrast, Chlenski et al. (2024) proposes HYPERDT taking a hyperplane-based perspective on Euclidean decision tree learning and following up with a modification of the splitting criterion to accommodate axis-inclined hyperplane splits.

By exploiting the (almost-Euclidean) behavior of hyperplanes in the ambient space of the Lorentz model of hyperbolic space, HYPERDT is able to more closely parallel Euclidean tree learning. In particular, it is able to consider each spacelike dimension separately and has identical training and inference complexity to its Euclidean counterparts. However, its modified, hyperplane-based splitting criteria are no longer compatible with the threshold-based decision tree algorithms, necessitating its own ad hoc implementation. This implementation cannot make use of the many optimizations made to the speed and accuracy of existing decision tree methods.

In this paper, we rewrite HYPERDT as Fast-HYPERDT, relying on the Beltrami-Klein model of hyperbolic space. This representation allows geodesic decision boundaries to become (axis-parallel) Euclidean hyperplanes, recovering a thresholding-based variant of HYPERDT. Fast-HYPERDT unlocks three advantages:

1. Leveraging optimized implementations is thousands of times faster than existing HYPERDT;

2. It is intrinsically compatible with well-known Euclidean methods like SCIKIT-LEARN; and

3. It easily extends to other tree-based paradigms, such as oblique decision trees or XGBOOST.

## 1.1 Related work

**Non-Euclidean Decision Trees.** This work is most closely related to other works proposing the use of tree-based models in non-Euclidean spaces. Chlenski et al. (2024) is the starting point for our model; Chlenski et al. (2025a) extends decision trees to (products of) any constant-curvature space; Doorenbos et al. (2023) offers a complementary, horosphere-based perspective on random forests in hyperbolic space. Tsagkrasoulis & Montana (2017) proposes a method to fit random forests for non-Euclidean *labels*, while (Bauerschmidt et al., 2021) explores connections between random spanning forests and hyperbolic symmetry. More recently, Suganthan et al. (2025) explores generalizing XGBoost to the Poincaré Ball.

**Hyperbolic Classifiers.** The problem of classification and regression in hyperbolic spaces has received a lot of attention in the literature. Many approaches involve modifying neural networks to work with hyperbolic data (Chami et al., 2019; Ganea et al., 2018; Chen et al., 2022; Bdeir et al., 2023; Khan et al., 2024). We are indebted to the literature on hyperbolic SVMs (Cho et al., 2018; Fan et al., 2023); the former makes use of the Klein model as well. Regression on hyperbolic manifolds has been studied by Marconi et al. (2020).

**Machine Learning in the Beltrami-Klein Model.** Mao et al. (2024) extends the Klein model for neural nets and provides an excellent overview of the state of the literature, including applications of the Klein model for graph embeddings (McDonald & He, 2019; Yang et al., 2023), protein sequences (Ali et al., 2024), minimal spanning trees (García-Castellanos et al., 2025), and scene images (Bi et al., 2017). We highlight Nielsen & Nock (2010), an early work using the Klein model to find Voronoi diagrams in hyperbolic space, whose discussion of partitions of the Beltrami-Klein model was foundational for our work.

## 2 Preliminaries

**Notation.** Angle brackets $\langle \cdot, \cdot \rangle$ and double bars $\| \cdot \|$ denote the Euclidean inner product and norm; the Lorentz form is $\langle \cdot, \cdot \rangle_{\mathbb{L}}$. Unless subscripted, $\langle \cdot, \cdot \rangle$ and $\| \cdot \|$ are Euclidean. We write $\mathbb{B}$ (Beltrami–Klein) and $\mathbb{L}$ (Lorentz) for the hyperbolic models (dimension $d$ and curvature $K$ understood unless stated). Where necessary, we use $\mathbb{B}$ and $\mathbb{L}$ as subscripts to make it clear which model is being referred to: in particular, $\delta_{\mathbb{B}}$ refers to the distance function in the Beltrami-Klein model, and $\delta_{\mathbb{L}}$ refers to the distance function in the Lorentz model. We let $\mathbf{u}, \mathbf{v}$ denote elements of $\mathbb{L}$, whereas $\mathbf{p}, \mathbf{q}$ denote elements of $\mathbb{B}$. Where appropriate, we relate $\mathbf{u}, \mathbf{v}$ and $\mathbf{p}, \mathbf{q}$ explicitly through projections. Finally, $\mathbf{e}_i$ refers to a canonical basis vector, i.e. a vector that is one-hot in dimension $i$.

### 2.1 Hyperbolic Space

#### 2.1.1 The Lorentz Model of Hyperbolic Space

The Lorentz Model with sectional curvature $-K$ in $d$ dimensions, $\mathbb{L}_K^d$, is embedded inside Minkowski space, a vector space in $\mathbb{R}^{d+1}$ equipped with the Minkowski inner product:

$$\langle \mathbf{u}, \mathbf{v} \rangle_{\mathbb{L}} = -u_0 v_0 + \sum_{i=1}^{d} u_i v_i. \tag{1}$$

Points in the Lorentz Model are constrained to lie on the upper sheet of a two-sheeted hyperboloid with a constant Minkowski inner product $-1/K$:

$$\mathbb{L}_K^d = \left\{ \mathbf{u} \in \mathbb{R}^{d+1} : \langle \mathbf{u}, \mathbf{u} \rangle_{\mathbb{L}} = -\frac{1}{K}, \ u_0 > 0 \right\}. \tag{2}$$

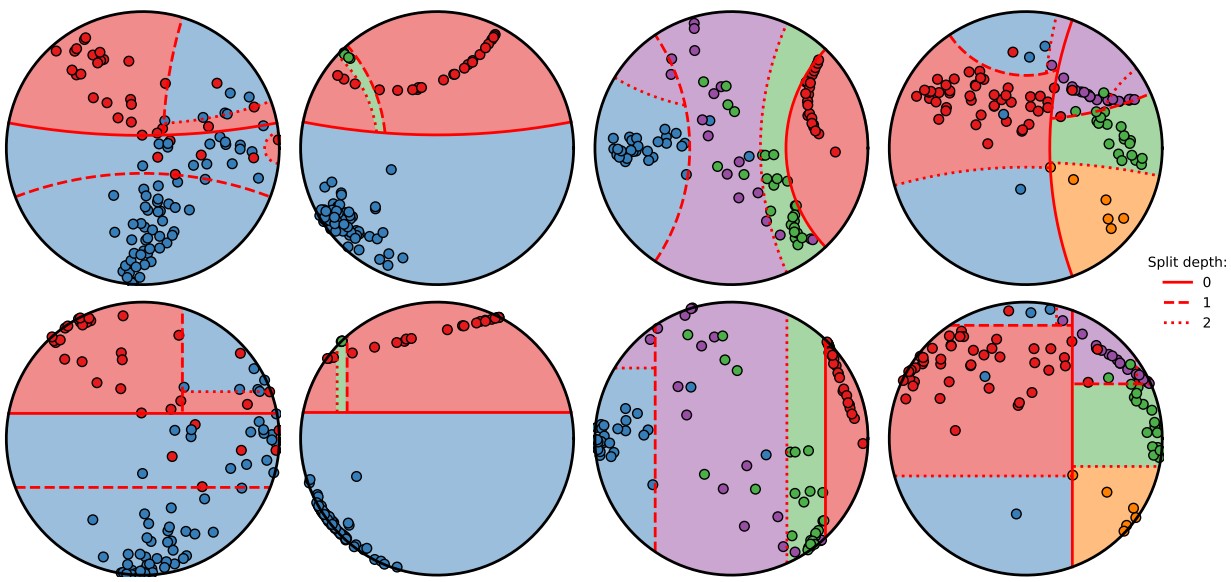

Figure 1: Original HYPERDT decision boundaries visualized in the Poincaré model (top) and in the Beltrami-Klein model (bottom). The top half of the figure is a reproduction of Figure 2 in Chlenski et al. (2024).

Distances between two points $\mathbf{u}, \mathbf{v} \in \mathbb{L}$, called geodesic distances, are computed as

$$\delta_{\mathbb{L}}(\mathbf{u}, \mathbf{v}) = \frac{1}{\sqrt{K}} \operatorname{arccosh}\left(-K \langle \mathbf{u}, \mathbf{v} \rangle_{\mathbb{L}}\right). \tag{3}$$

### 2.1.2 The Beltrami-Klein Model of Hyperbolic Space

The Beltrami-Klein Model realizes hyperbolic space as the open unit ball:

$$\mathbb{B}^d_K = \left\{\mathbf{p} \in \mathbb{R}^d : \|\mathbf{p}\| < 1\right\}. \tag{4}$$

$\mathbb{L}^d_K$ and $\mathbb{B}^d_K$ are related through the gnomonic projection $\phi_K : \mathbb{L}^d_K \to \mathbb{B}^d_K$ and its inverse $\phi_K^{-1} : \mathbb{B}^d_K \to \mathbb{L}^d_K$:

$$\phi_K(u_0, u_1, \ldots, u_d) = \left(\frac{u_1}{u_0}, \frac{u_2}{u_0}, \ldots, \frac{u_d}{u_0}\right) \tag{5}$$

$$\phi_K^{-1}(p_1, \ldots p_d) = \frac{1}{\sqrt{K}}\left(\frac{1}{\sqrt{1 - \|\mathbf{p}\|^2}}, \frac{p_1}{\sqrt{1 - \|\mathbf{p}\|^2}}, \ldots, \frac{p_d}{\sqrt{1 - \|\mathbf{p}\|^2}}\right). \tag{6}$$

(In Eqs 4 and 6, $\langle \cdot, \cdot \rangle$ and $\| \cdot \|$ are the Euclidean inner product and norm, respectively.) Indices for $\mathbf{u} \in \mathbb{L}^d_K$ start at 0, whereas the indices of $\mathbf{p} \in \mathbb{B}^d_K$ start at 1, reflecting the use of an extra dimension in the Lorentz model. This correspondence also sends hyperbolic geodesics to chords in $\mathbb{B}$; distances in $\mathbb{B}$ are defined by composing $\phi_K^{-1}$ with $\delta_{\mathbb{L}}$ as follows:

Let $\mathbf{a}, \mathbf{b} \in \mathbb{B}^d_K$ and set $\mathbf{x} = \phi_K^{-1}(\mathbf{a})$, $\mathbf{y} = \phi_K^{-1}(\mathbf{b})$. By (6),

$$\mathbf{x} = \frac{1}{\sqrt{K}} \frac{(1, \mathbf{a})}{\sqrt{1 - \|\mathbf{a}\|^2}}, \tag{7}$$

$$\mathbf{y} = \frac{1}{\sqrt{K}} \frac{(1, \mathbf{b})}{\sqrt{1 - \|\mathbf{b}\|^2}}. \tag{8}$$

Compute the Minkowski inner product:

$$
\begin{aligned}
\langle \mathbf{x}, \mathbf{y} \rangle_{\mathbb{L}} &= -x_0 y_0 + \sum_{i=1}^{d} x_i y_i \\
&= -\frac{1}{K} \frac{1}{\sqrt{1 - \|\mathbf{a}\|^2} \sqrt{1 - \|\mathbf{b}\|^2}} + \frac{1}{K} \frac{\langle \mathbf{a}, \mathbf{b} \rangle}{\sqrt{1 - \|\mathbf{a}\|^2} \sqrt{1 - \|\mathbf{b}\|^2}} \\
&= \frac{1}{K} \frac{-1 + \langle \mathbf{a}, \mathbf{b} \rangle}{\sqrt{1 - \|\mathbf{a}\|^2} \sqrt{1 - \|\mathbf{b}\|^2}} \\
\text{Thus,} \quad -K \langle \mathbf{x}, \mathbf{y} \rangle_{\mathbb{L}} &= \frac{1 - \langle \mathbf{a}, \mathbf{b} \rangle}{\sqrt{(1 - \|\mathbf{a}\|^2)(1 - \|\mathbf{b}\|^2)}}.
\end{aligned}
\tag{9}
$$

Composing with the Lorentz-model distance,

$$
\begin{aligned}
\delta_{\mathbb{B}}(\mathbf{a}, \mathbf{b}) &:= \delta_{\mathbb{L}}\big(\phi_K^{-1}(\mathbf{a}), \phi_K^{-1}(\mathbf{b})\big) \\
&= \frac{1}{\sqrt{K}} \operatorname{arccosh}\big(-K \langle \phi_K^{-1}(\mathbf{a}), \phi_K^{-1}(\mathbf{b}) \rangle_{\mathbb{L}}\big) \\
&= \frac{1}{\sqrt{K}} \operatorname{arccosh}\left( \frac{1 - \langle \mathbf{a}, \mathbf{b} \rangle}{\sqrt{(1 - \|\mathbf{a}\|^2)(1 - \|\mathbf{b}\|^2)}} \right).
\end{aligned}
\tag{10}
$$

An equivalent geometric construction for the geodesic distance is given in Appendix Section A.1.

**Einstein Midpoints.** A key advantage of the Beltrami-Klein model we will make use of in this paper is its closed-form formulation of geodesic midpoints. For $\mathbf{p}, \mathbf{q} \in \mathbb{B}$, the Einstein midpoint is given by

$$
m_{\mathbb{B}}(\mathbf{p}, \mathbf{q}) = \frac{\gamma_{\mathbf{p}} \mathbf{p} + \gamma_{\mathbf{q}} \mathbf{q}}{\gamma_{\mathbf{p}} + \gamma_{\mathbf{q}}}, \quad \text{where } \gamma_{\mathbf{x}} = \frac{1}{\sqrt{1 - \|\mathbf{x}\|^2}}.
\tag{11}
$$

Note that the conformal factor $\gamma_{\mathbf{x}}$ is proportional to the timelike dimension under the inverse gnomonic projection $\phi_K^{-1}$ given in Equation 6, scaled by a factor of $1/\sqrt{K}$. An alternative formulation for the Einstein midpoint, in which vectors in the Lorentz model are summed and re-normalized onto the manifold, is given in Appendix Section A.2.

## 2.2 Hyperbolic Hyperplanes

### 2.2.1 Hyperplanes

A (totally geodesic) hyperplane in $\mathbb{L}$ is the intersection

$$
H_{\mathbf{n}} = \big\{ \mathbf{u} \in \mathbb{L}_K^d : \langle \mathbf{u}, \mathbf{n} \rangle_{\mathbb{L}} = 0 \big\},
\tag{12}
$$

for a nonzero vector $\mathbf{n} = (a_0, \mathbf{a}) \in \mathbb{R}^{d+1}$. Under the Klein projection $\phi_K$, $H_{\mathbf{n}}$ maps to an affine Euclidean hyperplane

$$
\phi_K(H_{\mathbf{n}}) = \big\{ \mathbf{p} \in \mathbb{B}_K^d : \mathbf{a} \cdot \mathbf{p} = a_0 \big\},
\tag{13}
$$

since for $\mathbf{x} = (x_0, x_{1:d})$ we have $\mathbf{a} \cdot (x_{1:d}/x_0) = a_0$.

### 2.2.2 Nearest-point projection

The (signed) distance from a point $\mathbf{u} \in \mathbb{L}_K^d$ to $H_{\mathbf{n}}$ is given by

$$
\delta_{\mathbb{L}}(\mathbf{u}, H_{\mathbf{n}}) = \frac{1}{\sqrt{K}} \operatorname{arcsinh}\left( \frac{\sqrt{K} \langle \mathbf{u}, \mathbf{n} \rangle_{\mathbb{L}}}{\sqrt{\langle \mathbf{n}, \mathbf{n} \rangle_{\mathbb{L}}}} \right).
\tag{14}
$$

This formula highlights that the distance is fundamentally determined by the Lorentz inner product between the point and the hyperplane's normal vector. This is a direct consequence of the geometric interpretation of the Lorentzian time-like angle in the hyperboloid model (Ratcliffe, 2019, Thm.3.2.12).

If $H \subset \mathbb{L}_K^d$ is a nonempty closed totally geodesic hyperplane, then for every $\mathbf{u} \in \mathbb{L}_K^d$ there is a unique closest point $\pi_{H_\mathbf{n}}(\mathbf{u}) \in H_\mathbf{n}$, the geodesic from $\mathbf{u}$ to $\pi_{H_\mathbf{n}}(\mathbf{u})$ realizes $\delta_\mathbb{L}(\mathbf{u}, H_\mathbf{n})$ and meets $H_\mathbf{n}$ orthogonally (Bridson & Haefliger, 1999, Prop. II.2.4).

### 2.2.3 Consequence for HyperDT

HYPERDT uses *homogeneous* hyperplanes (ambient linear equation $\langle \mathbf{u}, \mathbf{n} \rangle_\mathbb{L} = 0$) whose normals are 2-sparse, i.e., supported on $\{0, i\}$ for some $i$. As established in Equation 14, the distance $\delta_\mathbb{L}(\mathbf{u}, H_\mathbf{n})$ is a direct function of the inner product $\langle \mathbf{u}, \mathbf{n} \rangle_\mathbb{L}$. For a 2-sparse normal, this inner product simplifies to $-a_0 u_0 + a_i u_i$. Consequently, for a fixed $H_\mathbf{n}$, the distance depends only on the pair $(u_0, u_i)$ (or, equivalently, on the Klein slope $u_i/u_0$). In other words, the coordinates $u_j$ with $j \notin \{0, i\}$ do not affect $\delta_\mathbb{L}(\mathbf{u}, H_\mathbf{n})$, so equidistance and midpoint computations can be carried out entirely in the relevant $(0, i)$ slice using the single scalar $\theta = \text{arccot}(u_i/u_0)$.

## 2.3 HyperDT

HYPERDT and HYPERRF, which were introduced in (Chlenski et al., 2024), are generalizations of Decision Trees (Classification and Regression Trees, (Breiman, 2017)) and Random Forests (Breiman, 2001) when $\mathbf{X} \in \mathbb{L}_K^{n \times d}$, $\mathbf{y} \in \mathbb{R}^n$.

### 2.3.1 Geodesically-Convex Splits

The central contribution of the HYPERDT algorithm is to observe that splits in conventional decision trees can be thought of in terms of dot products with the normal of a separating axis-parallel hyperplane:

$$S(\mathbf{x}, i, t) = \mathbf{1}_{x_i > t} = \text{Sign}(\mathbf{x} \cdot \mathbf{e}_i - t)^+, \tag{15}$$

where $t$ can be thought of as a bias term inducing a translation of the decision boundary by $t$ away from the origin along $\mathbf{e}_i$. This observation naturally allows us to consider a more generic split function

$$S(\mathbf{x}, \mathbf{n}, t) = \text{Sign}(\mathbf{x} \cdot \mathbf{n} - t)^+, \tag{16}$$

where $\mathbf{n} \in \mathbb{R}^{d+1}$ is the normal vector for any separating hyperplane, which we will call $H$. In HYPERDT, the set of candidate separating hyperplanes is restricted to those whose normal vectors take the form

$$\mathbf{n}(i, \theta) = (-\cos(\theta),\ 0,\ \ldots,\ 0,\ \sin(\theta),\ \ldots,\ 0), \tag{17}$$

where dimensions 0 and $i$ are the only nonzero dimensions, and where $t = 0$, ultimately yielding the HYPERDT splitting criterion:

$$S(\mathbf{x}, i, \theta) = \text{Sign}(\mathbf{x} \cdot \mathbf{n}(i, \theta))^+ = \text{Sign}(\mathbf{x}_i \sin(\theta) - x_0 \cos(\theta))^+. \tag{18}$$

Removing the bias term ensures geodesic convexity, as only homogeneous hyperplanes (i.e. hyperplanes containing the origin) have geodesically convex intersections with $\mathbb{L}$; on the other hand, parameterizing splits in terms of a spacelike dimension $i$ and an angle $\theta$ allows HYPERDT to retain the expressiveness and time complexity of traditional decision tree algorithms.

### 2.3.2 Hyperbolic Angular Midpoints

In CART, if a split falls between $\mathbf{u}, \mathbf{v} \in \mathbf{X}_{\text{train}}$, then the boundary is conventionally placed directly in between $\mathbf{u}$ and $\mathbf{v}$. That is, for a split in dimension $i$, we set the threshold using a simple arithmetic mean. In Euclidean space, this has the property that $\mathbf{u}$ and $\mathbf{v}$ are now equidistant from the separating hyperplane through $h_i = t$, which is a reasonable inductive bias for choosing where to separate two classes.

In hyperbolic space, the naive midpoint is likewise the angular bisector

$$\theta_{\text{naive}} = \frac{\theta_1 + \theta_2}{2}, \tag{19}$$

but in this case the hyperplanes parameterized by $h_0 \cos(\theta_1) = h_d \sin(\theta_1)$ and $h_0 \cos(\theta_2) = h_d \sin(\theta_2)$ would not intersect $\mathbb{L}$ at points that are equidistant to the hyperplane $h_0 \cos(\theta_{\text{naive}}) = h_d \sin(\theta_{\text{naive}})$ under the hyperbolic distance metric $\delta_{\mathbb{L}}$. In order to ensure equidistance, HYPERDT instead uses a more complex midpoint formula derived from $\delta_{\mathbb{L}}$:

$$m_{\mathbb{L}}(\theta_1, \theta_2) = \text{arccot}(\alpha + \beta\sqrt{\alpha^2 - 1}), \text{ where} \tag{20}$$

$$\alpha = \frac{\sin(2\theta_1 - 2\theta_2)}{2\sin(\theta_1 + \theta_2)\sin(\theta_2 - \theta_1)}, \tag{21}$$

$$\beta = \text{Sign}(\theta_1 + \theta_2 - \pi). \tag{22}$$

## 3 Speeding Up HyperDT

The algorithm for speeding up HYPERDT training (see Algorithm 1) involves three key steps:

1. **Preprocessing:** To preprocess $\mathbf{X} \in \mathbb{L}^{n \times d}$ for compatibility with Euclidean classifiers, it suffices to project $\mathbf{X}$ to the Beltrami-Klein model $\mathbf{X}_{\mathbb{B}} \in \mathbb{B}^{n \times d}$ using $\phi_K$ as defined in Equation 5.

---

**Algorithm 1:** Fast-HYPERDT training

**Input:** Dataset $\mathbf{X} \in \mathbb{L}_K^{n \times d}$, labels $\mathbf{y} \in \mathbb{R}^n$, curvature $K < 0$
**Output:** Hyperbolic Random Forest $\mathcal{R} = \{\mathcal{T}\}_{i=1}^{n_{\text{trees}}}$ (where $n_{\text{trees}} = 1$ for a single tree)

1 **Function** Preprocess($\mathbf{X}, K$):
2      **return** $\phi_K(\mathbf{X})$                                      `// Project to Klein model`

3 **Function** AdjustThresholds($\mathcal{T}, \mathbf{X}, K$):
4      **if** $\mathcal{T}$ *is a leaf* **then**
5          **return**
6      $d \leftarrow \mathcal{T}.\text{feature}$
7      $t \leftarrow \mathcal{T}.\text{threshold}$
8      $\mathbf{X} \leftarrow \mathbf{X}[\mathcal{T}.\text{subsample\_indices}]$      `// Random forests may use bootstrapped subsamples of X`
9      $\mathbf{X}^+ \leftarrow \{\mathbf{x} \in \mathbf{X} : x_d > t\}$
10     $\mathbf{X}^- \leftarrow \{\mathbf{x} \in \mathbf{X} : x_d \leq t\}$
11     $L \leftarrow \max_{\mathbf{x} \in \mathbf{X}^-}\{x_d\}$
12     $R \leftarrow \min_{\mathbf{x} \in \mathbf{X}^+}\{x_d\}$
13     node.threshold $\leftarrow$ EinsteinMidpoint($L, R, K$)                `// Uses Equation 11`
14     AdjustThresholds($\mathcal{T}.\text{left}, \mathbf{X}^-$)
15     AdjustThresholds($\mathcal{T}.\text{right}, \mathbf{X}^+$)

16 **Function** Postprocess($\mathcal{R}, \mathbf{X}, K$):
17     **for** $\mathcal{T} \in \mathcal{R}$ **do**
18        AdjustThresholds($\mathcal{T}, \mathbf{X}, K, \mathcal{T}$)
19     **return** $\mathcal{R}$

20 **Function** Train($\mathbf{X}_{\mathbb{L}}, \mathbf{y}, K$):
21     $\mathbf{X}_{\mathbb{B}} \leftarrow$ Preprocess($\mathbf{X}_{\mathbb{L}}, K$)
22     $\mathcal{R}_{\mathbb{E}} \leftarrow$ TrainEuclideanModel($\mathbf{X}_{\mathbb{B}}, \mathbf{y}$)       `// For instance, DecisionTreeClassifier.fit()`
23     $\mathcal{R}_{\mathbb{B}} \leftarrow$ Postprocess($\mathcal{R}_{\mathbb{E}}, \mathbf{X}_{\mathbb{B}}, K$)
24     **return** $\mathcal{R}_{\mathbb{B}}$

---

---

**Algorithm 2:** Fast-HYPERDT inference ($O(nh)$ version)

---

**Input:** Dataset $\mathbf{X} \in \mathbb{L}_K^{n \times d}$, trained model $\mathcal{R}_{\mathbb{B}}$
**Output:** Predictions $\hat{\mathbf{y}} \in \mathbb{R}^d$

**1** **Function** PredictNode($\mathbf{x}, \mathcal{T}$):
**2**    **if** $\mathcal{T}$ *is a leaf* **then**
**3**      **return** $\mathcal{T}$.label
**4**    $d \leftarrow \mathcal{T}$.feature
**5**    $t \leftarrow \mathcal{T}$.threshold
**6**    $\phi(\mathbf{x})_d \leftarrow x_d/x_0$              // Compute Klein model coordinates selectively
**7**    **if** $\phi(\mathbf{x})_d \leq t$ **then**
**8**      **return** PredictNode($\mathbf{x}, \mathcal{T}$.left)
**9**    **else**
**10**      **return** PredictNode($\mathbf{x}, \mathcal{T}$.right)

**11** **Function** Predict($\mathbf{X}_{\mathbb{L}}, \mathcal{R}_{\mathbb{B}}$):
**12**    $\hat{\mathbf{y}} \leftarrow$ array of size $|\mathbf{X}_{\mathbb{L}}|$
**13**    **for** $i \leftarrow 1$ **to** $n$ **do**
**14**      V $\leftarrow \{\}$                    // Initialize empty multiset
**15**      **for** $\mathcal{T} \in |\mathcal{R}_{\mathbb{B}}|$ **do**
**16**        $v \leftarrow$ PredictNode($\mathbf{X}_{\mathbb{B}}[i], \mathcal{T}$)
**17**        V $\leftarrow V \cup \{v\}$            // Add prediction to multiset
**18**      $\hat{\mathbf{y}}[i] \leftarrow$ Resolve($V$)     // However your ensemble aggregates tree votes
**19**    **return** $\hat{\mathbf{y}}$

---

**Algorithm 3:** Fast-HYPERDT inference (simple $O(n(d+h))$ version)

---

**Input:** Dataset $\mathbf{X} \in \mathbb{L}_K^{n \times d}$, curvature $K < 0$, trained model $\mathcal{R}_{\mathbb{B}}$
**Output:** Predictions $\hat{\mathbf{y}} \in \mathbb{R}^d$

**1** **Function** Predict($\mathbf{X}_{\mathbb{L}}, K, \mathcal{R}_{\mathbb{B}}$):
**2**    $\mathbf{X}_{\mathbb{B}} \leftarrow$ Preprocess($\mathbf{X}_{\mathbb{L}}, K$)
**3**    **return** PredictEuclidean($\mathbf{X}_{\mathbb{B}}, \mathcal{R}_{\mathbb{B}}$)     // For instance, DecisionTreeClassifier.predict()

---

2. **Train using Scikit-Learn:** Train a SCIKIT-LEARN-compatible Decision Tree or Random Forest on the transformed data $\mathbf{X}_{\mathbb{B}}$, yielding a trained Euclidean predictor $\mathcal{R}$.

3. **Postprocess:** Recompute geodesic midpoints for all decision boundaries in $\mathcal{R}$ and store them in the corrected hyperbolic predictor $\mathcal{R}_{\mathbb{B}}$. Given two points $\mathbf{u}, \mathbf{v} \in \mathbb{B}$, the corrected midpoint is simply the Einstein midpoint along the line connecting $\mathbf{u}$ and $\mathbf{v}$, as in Equation 11. This modification can be applied recursively to a trained tree, taking $u$ and $v$ as the closest values to either side of the learned threshold.

This approach allows us to leverage efficient Euclidean implementations of decision trees and random forests (for instance, SCIKIT-LEARN's RandomForestClassifier class) while maintaining the correct geometry of hyperbolic space with curvature $K$.

To speed up inference, we propose Algorithm 2, which preserves the asymptotic complexity of CART and HyperDT (see Theorem 4.6). For most practical purposes, it is generally faster to preprocess and use the built-in prediction functionality of the base model instead, as in Algorithm 3.

### 3.1 Extensions

In this section, we generalize Fast-HYPERDT to the Poincaré ball and extend it to use XGBoost base models. In Appendix Section B we extend Fast-HYPERDT to LightGBM and Oblique Decision Tree base models.

#### 3.1.1 Poincaré Ball Model

It is common for hyperbolic machine learning approaches to use the Poincaré ball model (which we will denote $\mathbb{P}_K$) rather than the hyperboloid or Beltrami-Klein models. We omit a detailed discussion of the properties of the Poincaré ball model, except to mention that it is possible to apply our approach to $\mathbf{X}_P \in \mathbb{P}_K^{n,d}$ using a projection $\rho_K : \mathbb{P}_K^d \to \mathbb{B}_K^d$. We include its inverse for completeness.

$$\rho_K(u_1, \ldots, u_d) = \left( \frac{u_1}{1 + \sqrt{1 - K\|\mathbf{u}\|^2}}, \; \frac{u_2}{1 + \sqrt{1 - K\|\mathbf{u}\|^2}}, \; \cdots, \; \frac{u_d}{1 + \sqrt{1 - K\|\mathbf{u}\|^2}} \right) \tag{23}$$

$$\rho_K^{-1}(v_1, \ldots, v_d) = \left( \frac{2v_1}{1 - K\|\mathbf{v}\|^2}, \; \frac{2v_2}{1 - K\|\mathbf{v}\|^2}, \; \cdots, \; \frac{2v_d}{1 - K\|\mathbf{v}\|^2} \right) \tag{24}$$

Once the points are projected to the Beltrami-Klein model, the rest of the computations proceed exactly as in the hyperboloid case.

All HyperDT classifiers are initialized with an `input_geometry` hyperparameter, which could be one of `hyperboloid`, `klein`, or `poincare`, and the appropriate conversion is applied during preprocessing.

#### 3.1.2 XGBoost

The extension of our approach to XGBoost (Chen & Guestrin, 2016) follows naturally from the decision tree implementation, as XGBoost fundamentally relies on the same axis-parallel splitting mechanism. While XGBoost introduces gradient boosting, regularization, and sophisticated loss function optimization, the geometric interpretation of its decision boundaries remains unchanged. Each tree in the trained XGBoost ensemble can be postprocessed analogously to a SCIKIT-LEARN Random Forest.

The primary implementation challenge lies in accessing individual tree structures within the XGBoost model, as the internal representation differs from SCIKIT-LEARN's. Nevertheless, once the nodes are accessible, the threshold adjustment process remains identical, maintaining the correct hyperbolic geometry throughout the ensemble. We allow XGBoost as a valid backend in our implementation.

Because XGBoost, unlike SCIKIT-LEARN, does not store a record of sample indices used to train each individual tree, we introduce a `override_subsample` hyperparameter to our XGBoost model. By default, `override_subsample = True`, resetting the `subsample` hyperparameter of all XGBoost models to 1.0. If users set `override_subsample = False`, the XGBoost model trains using subsampling, but the postprocessing uses the entire training set. In this case, a warning is issued that postprocessing is approximate.

## 4 Theoretical Results

In this section, we establish the theoretical foundation for our Fast-HYPERDT algorithm by proving its equivalence to the original HYPERDT method while demonstrating its computational advantages. The key insight of our approach lies in exploiting the geometric properties of the Beltrami-Klein model, where geodesics appear as straight lines, to leverage efficient Euclidean decision tree implementations.

**Lemma 4.1** (Hyperplane Classification Equivalence). *Let* $\mathbf{n} = (-\cos\theta, 0, ..., 0, \sin\theta, 0, ...) \in \mathbb{R}^{d+1}$ *be a normal vector for a hyperplane. For any* $\mathbf{x} \in \mathbb{L}_K^d$, *the HYPERDT decision rule satisfies:*

$$\mathrm{Sign}(\langle \mathbf{x}, \mathbf{n} \rangle) = \mathrm{Sign}\left( \phi_K(\mathbf{x})_i - \cot(\theta) \right). \tag{25}$$

*Proof.*

$$\langle \mathbf{x}, \mathbf{n} \rangle > 0 \iff -x_0 \cos(\theta) + x_i \sin(\theta) > 0 \iff \frac{x_i}{x_0} > \frac{\cos(\theta)}{\sin(\theta)} \iff \phi_K(\mathbf{x})_i > \cot(\theta). \tag{26}$$

Since $x_0 > 0$ in the Lorentz model, dividing by $x_0$ never affects the direction of this inequality. $\qquad\square$

**Lemma 4.2** (Threshold Invariance). *For any decision tree trained to optimize an information gain objective, the feature space partitioning is invariant to the choice of threshold placement between adjacent feature values. Specifically, if $u < v$ are consecutive observed values of a feature, then any threshold $t \in (u, v)$ produces identical tree structures and decision boundaries.*

*Proof.* Let us define the Information Gain (IG) for splitting a dataset $(\mathbf{X}, \mathbf{y})$ using threshold $t$ on feature $i$:

$$\text{IG}(\mathbf{y}, t) = H(\mathbf{y}) - \left( \frac{|\mathbf{y}^+|}{|\mathbf{y}|} H(\mathbf{y}^+) + \frac{|\mathbf{y}^-|}{|\mathbf{y}|} H(\mathbf{y}^-) \right), \text{ where}$$
$$\mathbf{y}^+ = \{y_j \in \mathbf{y} \ : \ (x_j)_i \geq t\}, \quad \mathbf{y}^- = \{y_j \in \mathbf{y} \ : \ (x_j)_i < t\}. \tag{27}$$

Here, $H(\cdot)$ denotes some impurity measure, e.g. Gini impurity. Now, for some values $u < v$ such that no $\mathbf{x} \in \mathbf{X}$ has a value between $u$ and $v$ at feature $i$, then the partition of $\mathbf{y}$ into $\mathbf{y}^+$ and $\mathbf{y}^-$ remains identical for any choice of $t \in (u, v)$.

Because information gain depends solely on this partition and not on the specific threshold value, the IG value remains constant for all $t \in (u, v)$. Consequently, any such threshold produces identical tree structures and decision boundaries when optimizing an information gain objective. $\qquad\square$

**Lemma 4.3** (Midpoint Equivalence). *Let $\mathbf{u}, \mathbf{v} \in \mathbb{L}_K^d$ and define $\theta_x := \text{arccot}(x_d/x_0)$ for any $x = (x_0, \ldots, x_d, \ldots) \in \mathbb{L}_K^d$. Let $\phi : \mathbb{L}_K^d \to \mathbb{B}_K^d$ be the gnomonic projection, so that $[\phi(x)]_d = x_d/x_0$. Write $\mathbf{p} = \phi(\mathbf{u})$ and $\mathbf{q} = \phi(\mathbf{v})$, and let $m_\mathbb{B}(\mathbf{p}, \mathbf{q})$ be the Einstein midpoint in $\mathbb{B}$ from 11. Define*

$$m_\mathbb{L} := \phi_K^{-1} \left( m_\mathbb{B}(\mathbf{p}, \mathbf{q}) \right) = \frac{\mathbf{u} + \mathbf{v}}{\sqrt{-K \langle \mathbf{u} + \mathbf{v}, \mathbf{u} + \mathbf{v} \rangle_\mathbb{L}}} \in \mathbb{L}_K^d, \tag{28}$$

*using 35. Then the hyperbolic angular midpoint $\theta_\mathbb{L}(\theta_\mathbf{u}, \theta_\mathbf{v})$ from 22 satisfies*

$$\theta_\mathbb{L}(\theta_\mathbf{u}, \theta_\mathbf{v}) = \theta_{m_\mathbb{L}} = \text{arccot}\left( [m_\mathbb{B}(\mathbf{p}, \mathbf{q})]_d \right). \tag{29}$$

*Proof.* The lemma asserts the identity of two different constructions for a midpoint angle. We will prove this by showing that the point $m_\mathbb{L}$, derived from the Einstein midpoint, satisfies the geometric definition of the point whose angle is given by the hyperbolic angular midpoint formula.

The hyperbolic angular midpoint, whose angle $\theta_\mathbb{L}(\theta_\mathbf{u}, \theta_\mathbf{v})$ is given by Eq. 22, is defined as the *unique point* on the geodesic connecting $\mathbf{u}$ and $\mathbf{v}$ that is equidistant from them. Hyperbolic space is geodesically convex, which guarantees the existence and uniqueness of such a point (Ratcliffe, 2019).

Our strategy is to prove algebraically that the point $m_\mathbb{L}$ from Eq. 28 also satisfies these two defining properties: (a) it lies on the geodesic between $\mathbf{u}$ and $\mathbf{v}$, and (b) it is equidistant from them.

By its definition in Eq. 28, $m_\mathbb{L}$ is a positive scalar multiple of the vector sum $\mathbf{u} + \mathbf{v}$. As a linear combination of $\mathbf{u}$ and $\mathbf{v}$, the point $m_\mathbb{L}$ must lie in the 2-dimensional plane spanned by $\mathbf{u}$ and $\mathbf{v}$. In the Lorentz model, the geodesic connecting two points is the intersection of the hyperboloid with the 2D plane they span. Therefore, $m_\mathbb{L}$ lies on the geodesic between $\mathbf{u}$ and $\mathbf{v}$.

To prove equidistance, $\delta_\mathbb{L}(m_\mathbb{L}, \mathbf{u}) = \delta_\mathbb{L}(m_\mathbb{L}, \mathbf{v})$, it is sufficient to show that the Lorentz inner products are equal, i.e., $\langle m_\mathbb{L}, \mathbf{u} \rangle_\mathbb{L} = \langle m_\mathbb{L}, \mathbf{v} \rangle_\mathbb{L}$. Let $C = \sqrt{-K \langle \mathbf{u} + \mathbf{v}, \mathbf{u} + \mathbf{v} \rangle_\mathbb{L}}$ be the normalization constant.

$$\langle m_\mathbb{L}, \mathbf{u} \rangle_\mathbb{L} = \left\langle \frac{\mathbf{u} + \mathbf{v}}{C}, \mathbf{u} \right\rangle_\mathbb{L} = \frac{1}{C}(\langle \mathbf{u}, \mathbf{u} \rangle_\mathbb{L} + \langle \mathbf{v}, \mathbf{u} \rangle_\mathbb{L})$$
$$\langle m_\mathbb{L}, \mathbf{v} \rangle_\mathbb{L} = \left\langle \frac{\mathbf{u} + \mathbf{v}}{C}, \mathbf{v} \right\rangle_\mathbb{L} = \frac{1}{C}(\langle \mathbf{u}, \mathbf{v} \rangle_\mathbb{L} + \langle \mathbf{v}, \mathbf{v} \rangle_\mathbb{L})$$

By the hyperboloid constraint (Eq. 2), we have $\langle \mathbf{u}, \mathbf{u} \rangle_\mathbb{L} = \langle \mathbf{v}, \mathbf{v} \rangle_\mathbb{L} = -1/K$. Therefore, the two inner products are identical, proving that $m_\mathbb{L}$ is equidistant from $\mathbf{u}$ and $\mathbf{v}$.

We have shown that $m_\mathbb{L}$ is the unique point on the geodesic between $\mathbf{u}$ and $\mathbf{v}$ that is equidistant from them. This is precisely the definition of the point whose angle is $\theta_\mathbb{L}(\theta_\mathbf{u}, \theta_\mathbf{v})$. Thus, the points are identical, and so are their angles: $\theta_\mathbb{L}(\theta_\mathbf{u}, \theta_\mathbf{v}) = \theta_{m_\mathbb{L}}$.

Finally, the angle $\theta_{m_\mathbb{L}}$ is given by $\mathrm{arccot}([m_\mathbb{L}]_d / [m_\mathbb{L}]_0)$. As established in the Preliminaries section (specifically Eq. 35 and its derivation), the ratio of components of $m_\mathbb{L}$ is identical to the $d$-th coordinate of the Klein-projected point $m_\mathbb{B}(\mathbf{p}, \mathbf{q})$.

$$\frac{[m_\mathbb{L}]_d}{[m_\mathbb{L}]_0} = [m_\mathbb{B}(\mathbf{p}, \mathbf{q})]_d$$

This directly yields the second equality: $\theta_{m_\mathbb{L}} = \mathrm{arccot}([m_\mathbb{B}(\mathbf{p}, \mathbf{q})]_d)$. This completes the proof of the full identity. $\square$

*Remark* 4.4. The three lemmas correspond directly to our algorithm's stages: (1) Klein projection enables Euclidean thresholds (Lemma 4.1; (2) Invariance permits off-the-shelf training (Lemma 4.2); and (3) Midpoint correction recovers hyperbolically equidistant decision boundaries (Lemma 4.3).

**Theorem 4.5** (Algorithmic Equivalence). *Assuming that ties in information gain never occur, or are handled identically by* HYPERDT *and Fast-*HYPERDT*, both methods produce identical decision boundaries. More precisely: letting $\mathcal{D}_H$ be the partition of $\mathbb{L}$ induced by* HYPERDT *and $\mathcal{D}_F$ be the partition of $\mathbb{B}$ induced by Fast-*HYPERDT *on the same dataset, $\phi(\mathcal{D}_H) \equiv \mathcal{D}_F$.*

*Proof.* We proceed by induction on the depth of a split.

**Base case.** At depth 0, the manifold is unpartitioned: $\phi(\mathcal{D}_H) = \phi(\mathbb{L}) = \mathbb{B} = \mathcal{D}_F$.

**Inductive hypothesis.** Assume that we have $\phi(\mathcal{D}_H) = \mathcal{D}_F$ for all $D, F$ of maximum depth $d$.

**Inductive step.** By the inductive hypothesis, any region $R \subseteq \mathbb{L}$ being split at depth $d + 1$ is already equivalent via $\phi(\mathcal{D}_H) = \mathcal{D}_F$. By Lemma 4.1, partitions by equivalent hyperplanes induce equivalent splits, and by Lemma 4.2, both algorithms learn the same partition of the labels. Finally, by Lemma 4.3 the midpoints of the partitioned labels also coincide. $\square$

**Theorem 4.6** (Computational Complexity). *Let $n$ be the number of training points, $d$ the number of input features, and $h \leq \log n$ the height of each (balanced) tree. With presorted feature lists, Fast-*HYPERDT *trains in $O(nd \log n)$ time and performs inference in $O(hn)$ time, matching both Euclidean CART and* HYPERDT.

*Proof.* Let $t$ denote the number of trees in the ensemble (set $t = 1$ for a single tree). We analyze one tree and multiply by $t$ at the end.

**CART.** With presorted features, a standard Euclidean decision tree fits in $O(dn \log n)$ time and predicts in $O(hn)$ time (Sani et al., 2018). The level-order scan variant costs $O(dnh)$ to fit, but we quote the presorted bound for definiteness. Bounding $h \leq \log_2(n)$, a reasonable assumption for balanced trees, equates the two complexities.

**HyperDT.** HYPERDT differs from CART by a constant-time modification to the thresholding rule, so its core fitting cost is asymptotically that of CART.

**Preprocessing.** Applying $\phi_K(\cdot)$ to every sample touches each coordinate once, giving $O(nd)$.

**Postprocessing.** Each internal node (at most $2^h - 1 < 2n$) needs the two sample values nearest its threshold. Because the active sample set halves at every depth, a single linear pass suffices, so this stage is $O(n)$ per depth, giving $O(nh)$ total.

**Training.** Thus, the total training time for Fast-HYPERDT is

$$O(\text{train}) = O(dn) + O(dn \log n) + O(hn) = O(dn \log n) \tag{30}$$

under the assumption that $h \leq \log_2(n)$. Multiplying by $t$ yields $O(tdn \log n)$ for an ensemble.

**Inference.** Algorithm 2 visits exactly $h$ nodes per example and evaluates the projection at a single dimension for each visit, so prediction is $O(hn)$ per tree, or $O(thn)$ for the ensemble. $\square$

## 5 Experimental Results

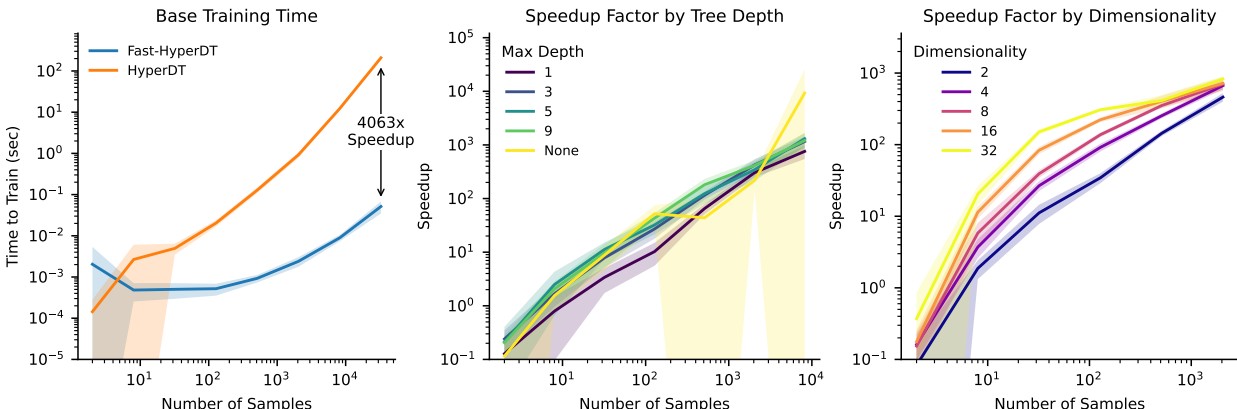

Figure 2: **Left:** We compare the training time of HyperDT and Fast-HyperDT across different training set sizes. Fast-HyperDT is consistently faster than HyperDT for 8 or more samples; for 32,768 samples, Fast-HyperDT trained on average 4,063× faster. **Middle:** We observe similar levels of speedup across different maximum depths, with some higher-variance behavior in the unlimited-depth regime. **Right:** We observe consistent speedups across dimensionalities, with higher-dimensional data speeding up more.

### 5.1 Experimental Setup

To match Scikit-Learn's tiebreaking behavior more closely, we modified HyperDT by changing the split criterion from $x < t$ to $x \leq t$ and reversing the order in which points are considered. We also set the random seed for Scikit-Learn decision trees to a constant value. This is necessary because, even when Scikit-Learn decision trees have subsampling disabled, they still randomly permute the features, which can affect the tiebreaking behavior of the algorithm.

Except where otherwise specified, we used trees with a maximum depth of 5, forests consisting of 100 trees, and no further restrictions on what splits are considered (e.g. minimum number of points in a leaf). We did not tune hyperparameters via cross-validation, as our focus is on demonstrating methodological equivalence and efficiency rather than squeezing out predictive performance.

We sample synthetic data from a mixture of wrapped Gaussians (Nagano et al., 2019), a common way of benchmarking hyperbolic classifiers (Cho et al., 2018; Chlenski et al., 2024). For all mixtures of Gaussians, we sampled the means from a wrapped normal distribution with a mean at the origin $\mu_0$ and covariance matrix equal to $\mathbf{I}/d$, where $d$ is the total number of dimensions; each point was then sampled from a wrapped normal distribution with a mean at its sampled origin $\mu$ and covariance matrix $\mathbf{I}/d$. Except where otherwise specified, we use a mixture of 8 Gaussians with class probabilities sampled from a uniform distribution. For regression datasets, we follow Chlenski et al. (2025b) in applying cluster-specific slopes and intercepts to the initial vectors sampled from Gaussian distribution to generate regression targets. All hyperbolic manifolds in our synthetic experiments use $K = 1$.

Benchmarks were conducted on an Ubuntu 22.04 machine equipped with an Intel Core i7-8700 CPU (6 cores, 3.20 GHz), an NVIDIA GeForce GTX 1080 GPU with 11 GiB of VRAM, and 15 GiB of RAM. Experiments were implemented using Python 3.11.0, accelerated by CUDA 11.4 with driver version 470.199.02. We used Scikit-Learn version 1.7.1, XGBoost 3.0.4, LightGBM 4.6.0, and scikit-obliquetree 0.1.4.

### 5.2 Agreement and Timing Benchmarks

To evaluate the time to train Fast-HyperDT, we trained HyperDT and Fast-HyperDT decision trees on varying numbers of samples from a mixture of wrapped Gaussian distributions. Figure 5 compares the training speeds of HyperDT and Fast-HyperDT on varying numbers of samples, revealing a 4,063

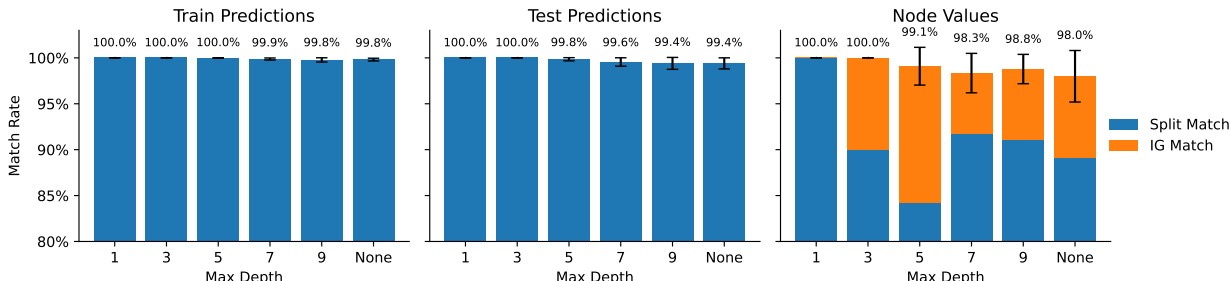

Figure 3: We compare the predictions and split values of HYPERDT and Fast-HYPERDT using a SCIKIT-LEARN backend. Error bars represent 95% confidence intervals. **Left:** The agreement between both models on the training data. **Middle:** The agreement between both models on the testing data. **Right:** Node-by-node split agreement, distinguishing between exact matches (blue) and matches that are equivalent in information gain (orange).

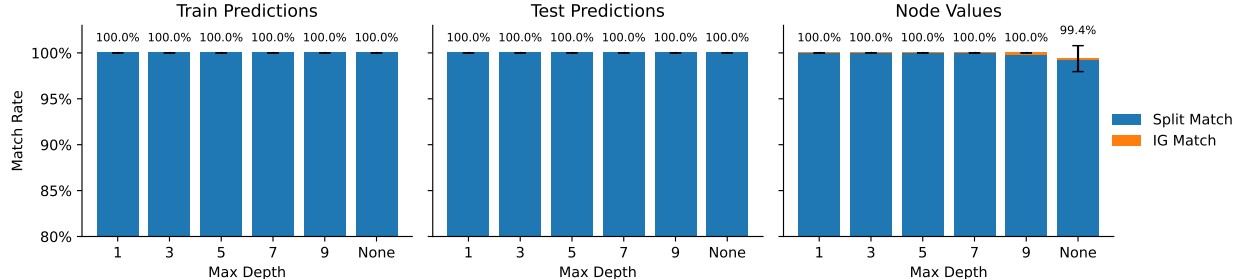

Figure 4: The results of Figure 3, compared to a deterministic baseline model designed to match HYPERDT tree-building logic exactly.

× speedup when training decision trees on 32,768 samples. Because Scikit-Learn's fit implementation is compiled in Cython and does not expose subcomponents, we cannot meaningfully attribute runtime to subroutines; instead we report end-to-end wall-clock times under varying number of samples, dimensionality, and maximum tree depth for Fast-HYPERDT.

Given the theoretical results in Section 4, it is natural to expect the predictions and splits of HYPERDT and Fast-HYPERDT to match exactly; instead, Figure 3 reveals a high but imperfect degree of correspondence between the two. We speculated that differences in pragmatic factors, such as tiebreaking rules (which split to prefer when two splits have the same information gain) and numerical precision, might be behind these slight differences in training behavior.

To investigate, we tested 10,000 different seeds comparing HYPERDT and Fast-HYPERDT on Gaussian mixtures of 1,000 points. Of these, we found that 9,934 splits matched exactly, while 43 splits matched in information gain (up to a tolerance of 0.0001). Figure 5 shows an example of information gains for 4 sets of features, revealing a tie between the best split along Features 1 and 2. Although we attempted to align the tiebreaking behavior of SCIKIT-LEARN and HYPERDT decision trees, we were never able to perfectly match the splits in all cases.

For the remaining 22 splits, Fast-HYPERDT always achieved a higher information gain than HYPERDT did, and angles tended to cluster near $\pi/4$ and $3\pi/4$. This seems to suggest that HYPERDT suffers from some numerical stability issues for extreme angles, which Fast-HYPERDT, likely by virtue of omitting the inverse tangent operation used to compute angles in the original HYPERDT algorithm, manages to avoid.

To close the performance gap, we implemented the exact tree-construction logic of HYPERDT, but using conventional thresholding-based split, and used this as a backend instead. The results for this baseline are showing in Figure 4: predictions were identical, and splits were identical almost everywhere. The remaining

Table 1: Parity with original HYPERDT/HYPERRF on the same simulated classification tasks. Accuries are (percentage ± 95% CI); Δ is Fast−Base as an absolute change in % for classification; "Prediction agreement" refers to the fraction of points for which HYPERDT and Fast-HYPERDT produced the same predictions. Despite implementation differences, we see almost identical performance between the HYPERDT and HYPERDT base tree implementations, and some divergence between their random forest variants.

| $d$ | Model | HYPERDT | Fast-HYPERDT | Δ | Prediction agreement |
|---|---|---|---|---|---|
| 2 | DT | 44.33 ± 1.57% | 44.33 ± 1.57% | +0.01 ± 0.02% | 99.98 ± 0.04% |
| 4 | DT | 37.81 ± 1.29% | 37.81 ± 1.29% | +0.00 ± 0.00% | 100.00 ± 0.00% |
| 8 | DT | 33.00 ± 1.21% | 32.99 ± 1.21% | -0.01 ± 0.01% | 99.92 ± 0.12% |
| 16 | DT | 29.07 ± 1.04% | 29.07 ± 1.04% | +0.00 ± 0.00% | 100.00 ± 0.00% |
| 32 | DT | 26.95 ± 1.17% | 26.95 ± 1.17% | +0.00 ± 0.00% | 100.00 ± 0.00% |
| 64 | DT | 25.04 ± 1.02% | 25.04 ± 1.02% | +0.01 ± 0.01% | 99.44 ± 1.06% |
| 2 | RF | 44.89 ± 1.61% | 45.60 ± 1.54% | +0.71 ± 0.81% | 78.93 ± 4.08% |
| 4 | RF | 37.78 ± 1.22% | 39.44 ± 1.35% | +1.67 ± 0.88% | 67.13 ± 3.98% |
| 8 | RF | 32.06 ± 1.19% | 33.91 ± 1.17% | +1.85 ± 0.75% | 56.91 ± 4.72% |
| 16 | RF | 28.53 ± 1.14% | 30.77 ± 1.03% | +2.24 ± 0.79% | 51.08 ± 4.26% |
| 32 | RF | 25.76 ± 1.10% | 27.56 ± 1.06% | +1.80 ± 0.75% | 40.15 ± 3.62% |
| 64 | RF | 23.69 ± 1.02% | 25.75 ± 1.07% | +2.06 ± 0.64% | 42.48 ± 3.75% |

differences all happen when splits fall close to the boundary of the Klein disk, pointing to numerical precision issues in the original HYPERDT's midpoint computation as a likely culprit.

Additionally, we benchmarked the change in accuracy across a series of 8-class classification benchmarks in Table 1 to evaluate the impact of switching between base HYPERDT and Fast-HYPERDT with a SCIKIT-LEARN base. We found that, despite slight differences in learned splits and predictions, the accuracy of HYPERDT and Fast-HYPERDT decision tree models was almost identical; on the other hand, SCIKIT-LEARN-based random forest models consistently outperformed their HYPERDT counterparts. This difference is likely driven by better randomization in the SCIKIT-LEARN implementation.

## 5.3 Other Models

In Table 2 we evaluate Fast-HYPERDT alongside a variety of other models on an 8-class classification task involving Gaussian mixtures. For each base class, we benchmark the base model on Lorentz coordinates (equivalent to not using Fast-HYPERDT at all), the base model with just the Fast-HYPERDT preprocessing step (conversion to Klein coordinates), and the full Fast-HYPERDT-wrapped model. In all cases, XGBoost models performed best, highlighting the value of extending this technique to hyperbolic space; moreover, we observe significant improvements from preprocessing points to move them from the Lorentz model to the Klein model, and modest improvements in accuracy from postprocessing them to update the midpoint.

This corroborates the midpoint ablation in Chlenski et al. (2024), which suggests that placing exact midpoints in the original HYPERDT algorithm offers a modest improvement over keeping the implementation with naive

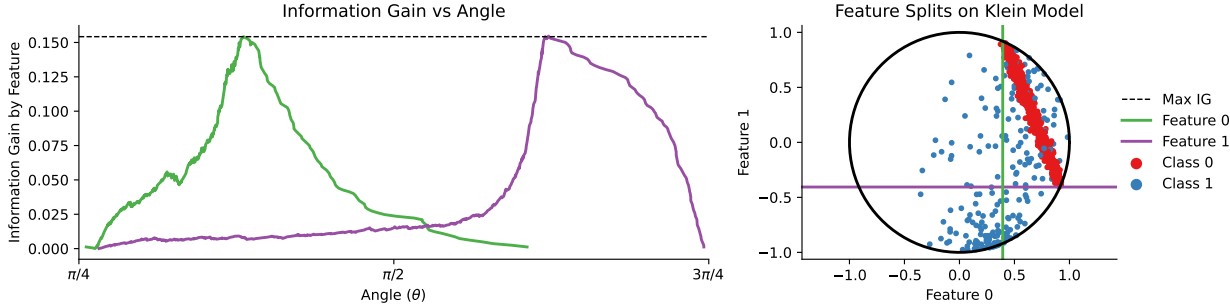

Figure 5: A Gaussian mixture dataset for which multiple splits attain the same information gain. **Left:** The information gains attained by each candidate split for each feature. **Right:** Each point in the Klein model, colored by class, and two splits on different features that attain the same information gain.

Table 2: Classification accuracies (in percent, ±95% CI) across 100 synthetic 8-class benchmarks using Gaussian mixture datasets of varying dimensionalities.

| | | Sklearn DT | | | Sklearn RF | | | XGBoost | | |
|---|---|---|---|---|---|---|---|---|---|---|
| Klein | | | ✓ | ✓ | | ✓ | ✓ | | ✓ | ✓ |
| Midpoint | | | | ✓ | | | ✓ | | | ✓ |
| Dimension | 2 | 53.5±1.7 | 56.5±1.7 | 56.5±1.7 | 56.1±1.6 | 60.1±1.6 | 60.1±1.6 | 55.3±2.1 | 57.4±1.9 | 57.4±1.9 |
| | 4 | 48.3±1.2 | 51.8±1.3 | 51.8±1.3 | 54.6±1.2 | 59.8±1.2 | 59.8±1.2 | 59.8±1.2 | 65.2±1.2 | 65.0±1.2 |
| | 8 | 42.8±1.2 | 44.2±1.3 | 44.2±1.3 | 53.2±1.0 | 56.9±1.1 | 57.1±1.0 | 63.4±0.9 | 69.3±0.8 | 69.3±0.9 |
| | 16 | 36.2±1.1 | 36.3±1.1 | 36.3±1.1 | 49.2±0.9 | 51.6±0.9 | 52.1±0.9 | 63.7±0.8 | 68.9±0.9 | 68.8±0.9 |
| | 32 | 30.9±1.1 | 31.1±1.1 | 31.1±1.1 | 43.5±1.1 | 46.5±1.0 | 46.4±1.0 | 58.8±1.0 | 63.5±1.0 | 63.5±1.0 |
| | 64 | 26.3±1.0 | 26.9±1.0 | 27.0±1.0 | 36.6±1.1 | 39.2±1.1 | 39.2±1.1 | 50.6±1.1 | 54.5±1.0 | 54.5±1.0 |
| | 128 | 24.5±1.1 | 24.9±1.0 | 24.9±1.0 | 31.7±1.2 | 33.7±1.2 | 33.7±1.2 | 42.4±1.4 | 46.1±1.2 | 46.1±1.2 |
| % Improv. | | — | +3.0% | +3.1% | — | +7.0% | +7.2% | — | +7.8% | +7.7% |

midpoints. We note that the two ablations actually coincide, as the angular bisector and the average of the Klein coordinates are equal.[1]

Complete benchmarks, which include results for oblique decision trees and LightGBM, can be found in Appendix Section C. Full classification results can be found in Table 4, and analogous results for regression can be found in Table 5.

### 5.4 WordNet classification

WordNet (Fellbaum, 2010) is a common real-world classification benchmark for hyperbolic embeddings. We evaluate our best-performing method, Fast-HYPERDT with XGBoost as its base classifier, alongside the HoroRF(Doorenbos et al., 2023) and EPBoost (Suganthan et al., 2025) classifiers, using the hyperparameters evaluated in Suganthan et al. (2025) and default hyperparameters for Fast-HYPERDT. Our results appear in Table 3: in general, our method is quite close to EPBoost in accuracy, beating this method 4 times, being beaten 3 times, and tying once.

Table 3: Micro-averaged F1 scores (in percent, ±95% CI) across 8 binarized classification benchmarks using WordNet embeddings.

| Benchmark | SVM | HoroRF | EPBoost | Fast-HYPERDT+XGBoost |
|---|---|---|---|---|
| animal | 99.00±0.15 | 98.96±0.14 | 99.42±0.13 | 99.43±0.15 |
| group | 94.13±0.19 | 94.12±4.20 | 97.42±0.17 | 97.47±0.17 |
| worker | 98.64±0.00 | 98.54±0.14 | 99.07±0.18 | 99.05±0.11 |
| mammal | 98.56±0.00 | 99.37±0.13 | 99.50±1.25 | 99.89±0.03 |
| tree | 98.76±0.00 | 98.65±0.18 | 99.22±0.11 | 99.21±0.07 |
| solid | 98.50±0.00 | 98.79±0.16 | 99.42±0.07 | 99.43±0.04 |
| occupation | 99.65±0.00 | 99.57±0.10 | 99.66±0.03 | 99.66±0.05 |
| rodent | 99.83±0.00 | 99.80±0.08 | 99.83±0.00 | 99.82±0.00 |

## 6 Conclusions

In this work, we proposed Fast-HYPERDT, which rewrites HYPERDT as a wrapper around Euclidean tree-based models using the Beltrami-Klein model of hyperbolic space. We prove our method is equivalent to HYPERDT while being simpler, faster, and more flexible. We also demonstrate the superior speed and extensibility of our method empirically, in particular noting that the XGBoost variant of Fast-HYPERDT is vastly more accurate than base HYPERDT. For practitioners interested in using standard libraries, working in Klein coordinates directly without Fast-HYPERDT can be a performant baseline; however, the postprocessing variant we propose is even more accurate in most settings.

Future work can focus on extending Fast-HYPERDT to other methods, such as Isolation Forests (Liu et al., 2008) and rotation forests (Bagnall et al., 2020), which would address the absence of privileged basis dimen-

---

[1]This can be shown by converting angles $\theta_u = \text{arccot}(u_d/u_0)$ and $\theta_v = \text{arccot}(v_d/v_0)$ to their bisector $\theta_m = (\theta_u + \theta_v)/2$, then mapping back to Klein coordinates as $\cot(\theta_m) = (u_d/x_0 + x_d/x_0)/2$.

sions (Elhage et al., 2023) in hyperbolic embedding methods; extending Fast-HYPERDT to hyperspherical data as in Chlenski et al. (2025a); and extending the connection between HYPERDT, and decision trees to neural networks, as in Aytekin (2022) or via the polytope lens (Black et al., 2022).

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

# A  Further geometric details

## A.1  Alternative Klein distance construction

Equivalently to Equation 10, $\delta_{\mathbb{B}}$ can be computed using the following geometric construction: if the chord through two interior points $\mathbf{b}$ and $\mathbf{c}$ meets the boundary of the Klein disk at points $\mathbf{a}$ and $\mathbf{d}$ (with the ordering $\mathbf{a}, \mathbf{b}, \mathbf{c}, \mathbf{d}$ along the chord), then the hyperbolic distance between $\mathbf{b}$ and $\mathbf{c}$ is given by

$$\delta_{\mathbb{B}}(\mathbf{b}, \mathbf{c}) = \frac{1}{2} \ln \left( \frac{|\mathbf{ac}||\mathbf{bd}|}{|\mathbf{ab}||\mathbf{cd}|} \right), \tag{31}$$

where $|\mathbf{ab}|$ denotes the Euclidean norm of the line segment connecting $\mathbf{a}$ and $\mathbf{b}$, and so on (Papadopoulos & Troyanov, 2009). See Figure 6 for a visual illustration of the relevant quantities.

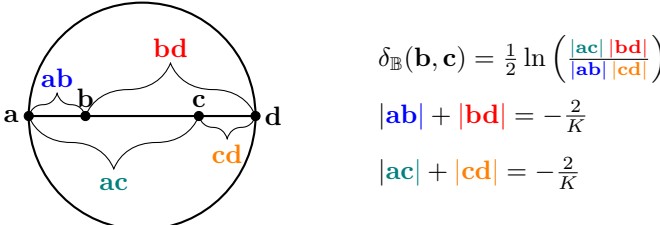

Figure 6:  The distance between nodes $\mathbf{b}$ and $\mathbf{c}$ in the Klein model can be understood through the cross-ratio between the line segments $\mathbf{ac}$ and $\mathbf{bd}$ with $\mathbf{ab}$ and $\mathbf{cd}$.

## A.2  Alternative Einstein midpoint formulation

Taking the Einstein midpoint in $\mathbb{B}$ is equivalent to summing the coordinates of two points in $\mathbb{L}$, then reprojecting onto $\mathbb{L}$, and finally converting to Klein coordinates: given $\mathbf{u}, \mathbf{v} \in \mathbb{L}_K^d$ and letting $\mathbf{p} = \phi_K(\mathbf{u}), \mathbf{q} = \phi_K(\mathbf{v})$,

$$\mathbf{u} + \mathbf{v} = \phi_K^{-1}(\mathbf{p}) + \phi_K^{-1}(\mathbf{q}) \tag{32}$$

$$= \frac{1}{\sqrt{K}} \left( (\gamma_{\mathbf{p}} + \gamma_{\mathbf{q}}), \gamma_{\mathbf{p}} \mathbf{p} + \gamma_{\mathbf{q}} \mathbf{q} \right) \tag{33}$$

$$= \frac{\gamma_{\mathbf{p}} + \gamma_{\mathbf{q}}}{\sqrt{K}} \left( 1, m_{\mathbb{B}}(\mathbf{p}, \mathbf{q}) \right) \tag{34}$$

And therefore $\dfrac{\mathbf{u} + \mathbf{v}}{\sqrt{-K \langle \mathbf{u} + \mathbf{v}, \mathbf{u} + \mathbf{v} \rangle_{\mathbb{L}}}} = \phi_K^{-1} \left( m_{\mathbb{B}}(\mathbf{p}, \mathbf{q}) \right). \tag{35}$

# B  Other models

## B.1  LightGBM

Similar to XGBoost, LightGBM (Ke et al., 2017) is a popular library for gradient-boosted decision trees. Also analogous to XGBoost, the innovations of LightGBM do not affect split geometry, so trained LightGBM models can be postprocessed as usual by editing thresholds in LightGBM's proprietary text-based model format. The subsampling issue also affects LightGBM models, so if users choose to set `bagging_fraction != 0`, the model also issues a warning and performs approximate postprocessing on the entire dataset.

## B.2  Oblique Decision Trees

For oblique decision trees, which use linear combinations of features rather than axis-parallel splits, our approach requires a natural extension to the projection mechanism. Instead of projecting onto basis dimension axes, we project data points onto the normal vector of each oblique hyperplane. Given a hyperplane defined

by $\mathbf{x} \cdot \mathbf{n} - t = 0$, we compute the projection of each point onto the normal vector $\mathbf{n}$. The threshold correction then proceeds identically to the axis-parallel case, applying the Einstein midpoint formula from Equation 11 to the scalar projections rather than to individual feature values.

This generalization maintains the geometric consistency of the decision boundaries in hyperbolic space while leveraging the increased flexibility of oblique splits. The implementation requires modifying the node structure to store the hyperplane parameters $\mathbf{w}$ and applying the projection during both training and inference phases, but the core principle of threshold adjustment remains unchanged.

Using this projection-based approach, we provide a hyperbolic variant of the HHCart (Wickrama-rachchi et al., 2015; 2019) and CO2 Forest (Norouzi et al., 2015) algorithms, as implemented in the `scikit-obliquetree` library (ECNU, 2021).

## C Detailed benchmark results

Table 4 contains full results for the classification benchmarks which are presented in an abridged version in Table 2. Table 5 contains results for regression benchmarks, which similarly show significant improvements from preprocessing, modest improvements from postprocessing, and a significant improvement from switching to an XGBoost base model. In general, we find robust improvements due to pre- and post-processing for SCIKIT-LEARN models (Decision Trees and Random Forests), as well as XGBoost; we also find that XGBoost models tend to outperform their counterparts. Oblique decision trees tend to significantly underperform other baselines. LightGBM performs well in base and base+preprocessing settings, but tends to underperform under postprocessing. This is likely due to differences in how trees are grown in LightGBM compared to other models.

Table 4: Classification scores (Accuracy percent, ±95% CI) across 100 synthetic classification datasets. Each classification benchmark seeks to classify points from a mixture of 8 Gaussians labeled according to the cluster from which they were drawn.

| | | Sklearn DT | | | Sklearn RF | | | XGBoost | | |
|---|---|---|---|---|---|---|---|---|---|---|
| **Klein** | | | ✓ | ✓ | | ✓ | ✓ | | ✓ | ✓ |
| **Midpoint** | | | | ✓ | | | ✓ | | | ✓ |
| Dimension | 2 | 53.5±1.7 | 56.5±1.7 | 56.5±1.7 | 56.1±1.6 | 60.1±1.6 | 60.1±1.6 | 55.3±2.1 | 57.4±1.9 | 57.4±1.9 |
| | 4 | 48.3±1.2 | 51.8±1.3 | 51.8±1.3 | 54.6±1.2 | 59.8±1.2 | 59.8±1.2 | 59.8±1.2 | 65.2±1.2 | 65.0±1.2 |
| | 8 | 42.8±1.2 | 44.2±1.3 | 44.2±1.3 | 53.2±1.0 | 56.9±1.1 | 57.1±1.0 | 63.4±0.9 | 69.3±0.8 | 69.3±0.9 |
| | 16 | 36.2±1.1 | 36.3±1.1 | 36.3±1.1 | 49.2±0.9 | 51.6±0.9 | 52.1±0.9 | 63.7±0.8 | 68.9±0.9 | 68.8±0.9 |
| | 32 | 30.9±1.1 | 31.1±1.1 | 31.1±1.1 | 43.5±1.1 | 46.5±1.0 | 46.4±1.0 | 58.8±1.0 | 63.5±1.0 | 63.5±0.9 |
| | 64 | 26.3±1.0 | 26.9±1.0 | 27.0±1.0 | 36.6±1.1 | 39.2±1.1 | 39.2±1.1 | 50.6±1.1 | 54.5±1.0 | 54.5±1.0 |
| | 128 | 24.5±1.1 | 24.9±1.0 | 24.9±1.0 | 31.7±1.2 | 33.7±1.2 | 33.7±1.2 | 42.4±1.4 | 46.1±1.2 | 46.1±1.2 |
| **% Improv.** | | — | +3.0% | +3.1% | — | +7.0% | +7.2% | — | +7.8% | +7.7% |

| | | CO2 | | | HouseHolderCART | | | LightGBM | | |
|---|---|---|---|---|---|---|---|---|---|---|
| **Klein** | | | ✓ | ✓ | | ✓ | ✓ | | ✓ | ✓ |
| **Midpoint** | | | | ✓ | | | ✓ | | | ✓ |
| Dimension | 2 | 18.3±1.7 | 19.7±2.0 | 20.0±1.9 | 12.4±1.3 | 13.0±1.4 | 13.0±1.4 | 55.8±2.0 | 58.4±1.9 | 36.8±1.7 |
| | 4 | 16.9±1.4 | 20.0±1.5 | 18.8±1.5 | 14.6±1.6 | 15.0±1.7 | 14.9±1.6 | 59.5±1.3 | 65.0±1.2 | 43.3±1.6 |
| | 8 | 15.0±1.3 | 17.4±1.3 | 17.4±1.3 | 12.4±1.4 | 12.5±1.3 | 12.5±1.3 | 63.2±1.0 | 68.0±0.9 | 53.3±1.3 |
| | 16 | 13.9±1.4 | 14.6±1.4 | 14.3±1.4 | 13.9±1.4 | 14.0±1.5 | 14.0±1.5 | 63.2±0.9 | 67.3±0.9 | 57.4±1.0 |
| | 32 | 12.3±1.2 | 12.3±1.4 | 12.5±1.4 | 12.9±1.3 | 12.7±1.3 | 12.8±1.3 | 58.8±0.9 | 62.2±1.0 | 54.5±1.1 |
| | 64 | 13.6±1.4 | 12.9±1.5 | 13.0±1.5 | 12.4±1.4 | 12.5±1.4 | 12.5±1.4 | 51.2±1.1 | 54.3±1.0 | 47.9±1.2 |
| | 128 | 13.7±1.3 | 13.6±1.5 | 13.8±1.5 | 13.0±1.4 | 13.0±1.4 | 13.0±1.4 | 43.6±1.2 | 46.3±1.2 | 40.6±1.3 |
| **% Improv.** | | — | +5.8% | +5.3% | — | +1.1% | +1.2% | — | +6.6% | -15.2% |

Table 5: Regression root mean squared error (RMSE, ±95% CI) scores across 100 synthetic regression datasets. Aside from using regression variants of all models, the benchmarking setup is otherwise identical to Table 4.

| Dimension | Sklearn DT | | | Sklearn RF | | | XGBoost | | |
|---|---|---|---|---|---|---|---|---|---|
| Klein | | ✓ | ✓ | | ✓ | ✓ | | ✓ | ✓ |
| Midpoint | | | ✓ | | | ✓ | | | ✓ |
| 2 | .204±.007 | .198±.008 | .198±.008 | .190±.007 | .186±.007 | .186±.007 | .208±.008 | .206±.008 | .206±.008 |
| 4 | .206±.007 | .196±.006 | .196±.006 | .188±.006 | .180±.005 | .180±.005 | .193±.006 | .182±.006 | .181±.006 |
| 8 | .202±.006 | .197±.006 | .197±.006 | .180±.005 | .175±.005 | .175±.005 | .180±.005 | .167±.005 | .167±.005 |
| 16 | .190±.006 | .184±.006 | .184±.006 | .164±.005 | .161±.005 | .161±.005 | .160±.005 | .151±.005 | .151±.005 |
| 32 | .184±.005 | .181±.004 | .181±.004 | .158±.004 | .156±.004 | .156±.004 | .155±.004 | .148±.004 | .148±.004 |
| 64 | .183±.004 | .180±.004 | .180±.004 | .157±.003 | .156±.003 | .156±.003 | .157±.003 | .153±.003 | .154±.003 |
| 128 | .177±.004 | .176±.003 | .176±.003 | .153±.003 | .152±.003 | .152±.003 | .156±.003 | .154±.003 | .154±.003 |
| % Change | — | -2.6% | -2.5% | — | -1.9% | -1.9% | — | -4.0% | -4.1% |

| Dimension | CO2 | | | HouseHolderCART | | | LightGBM | | |
|---|---|---|---|---|---|---|---|---|---|
| Klein | | ✓ | ✓ | | ✓ | ✓ | | ✓ | ✓ |
| Midpoint | | | ✓ | | | ✓ | | | ✓ |
| 2 | .200±.007 | .193±.007 | .194±.007 | .427±.008 | .422±.008 | .422±.008 | .192±.008 | .188±.007 | .225±.010 |
| 4 | .201±.006 | .191±.006 | .192±.006 | .445±.007 | .433±.007 | .433±.007 | .185±.006 | .175±.005 | .207±.007 |
| 8 | .196±.006 | .193±.006 | .192±.006 | .458±.006 | .450±.007 | .450±.007 | .173±.005 | .163±.005 | .184±.006 |
| 16 | .181±.005 | .180±.005 | .178±.005 | .461±.006 | .461±.006 | .461±.006 | .154±.005 | .146±.004 | .162±.005 |
| 32 | .176±.005 | .175±.005 | .174±.005 | .479±.005 | .476±.005 | .476±.005 | .148±.004 | .142±.003 | .154±.004 |
| 64 | .176±.004 | .175±.004 | .175±.004 | .492±.004 | .491±.004 | .491±.004 | .149±.003 | .146±.003 | .152±.004 |
| 128 | .174±.003 | .172±.003 | .171±.003 | .500±.004 | .501±.004 | .501±.004 | .148±.003 | .146±.003 | .149±.003 |
| % Change | — | -1.8% | -2.0% | — | -.9% | -.9% | — | -3.8% | +6.7% |

