# OpenReview forum: "Even Faster Hyperbolic Random Forests: A Beltrami-Klein Wrapper Approach"
_TMLR — Rejected by TMLR_

### Review · Reviewer_mwTC · 2025-07-20

**Summary Of Contributions:**

The paper introduces Fast-HyperDT, an optimized variant of the HyperDT algorithm for hyperbolic decision trees and random forests. Instead of directly working with the Lorentz model, it uses the Beltrami-Klein model to map hyperbolic geometry to Euclidean-like thresholding operations. This approach allows the use of standard, well-optimized decision tree frameworks (e.g., Scikit-Learn, XGBoost, LightGBM) with minimal modifications.

Key contributions:

1. Reformulation of HyperDT as a wrapper around existing Euclidean tree-based models via Klein projections.

2. Significant speed improvements.

3. Compatibility with mainstream libraries like Scikit-Learn, XGBoost, and LightGBM.

4. Theoretical equivalence to HyperDT via projection and midpoint corrections.

5. Provides empirical results on synthetic hyperbolic data.


Key Weaknesses:

1. Lack of direct comparison with original HyperDT on accuracy: The paper claims equivalence with HyperDT but does not report classification or regression accuracy comparisons with the original HyperDT on the same simulations.

2. No comparison to vanilla XGBoost/LightGBM: Since XGBoost and LightGBM are already highly optimized and accurate, it is unclear whether the hyperbolic wrapper actually adds value in accuracy or efficiency over these widely used baselines.

3. Tables 1 and 2 do not consider fair comparison: Accuracy improvements in these tables are primarily due to using boosted tree algorithms (e.g., XGBoost), not necessarily due to the hyperbolic method. Without vanilla XGBoost as a baseline, these comparison on accuracy are misleading.

4. Theory is descriptive rather than rigorous: The "theorems" (Section 4) are mostly equivalence arguments and complexity analyses. They are not formal mathematical theorems.

5. Evaluation limited to synthetic data, no real-world datasets.

**Audience:**

Yes

**Audience Explanation:**

While the contribution is relatively technical and algorithmic, it should still be of use, if the author can convincingly present the speed advantage and no accuracy loss.

**Claims And Evidence:**

No

**Claims Explanation:**

See weakness above. The speed improvement is partially justified but only on synthetic data. The accuracy claim is not fairly supported.

**Requested Changes:**

Suggested Improvements

1. Add direct accuracy comparisons with HyperDT on the same datasets to confirm the equivalence claim.

2. Include runtime and accuracy benchmarks against vanilla XGBoost and LightGBM (without the hyperbolic wrapper) to show whether the wrapper introduces overhead or accuracy gains.

3. Perform detailed simulation studies to isolate the impact of Klein projection and Einstein midpoint correction (e.g., comparing Scikit-Learn in Klein coordinates vs. Fast-HyperDT).

4. Expand experimental datasets to include real-world hyperbolic datasets for both speed and accuracy.

5. Refine theoretical contributions by formalizing equivalence proofs and quantifying numerical approximation errors (e.g., due to midpoint adjustments).

6. Clarify experimental tables: Clearly separate methods that are core contributions (Fast-HyperDT, Fast-HyperRF) from unrelated baselines (CO2, HHC, vanilla XGBoost).

Overall, the most important part to me is that authors should convincingly show that their methods provide significant speedup, while not losing accuracy versus the vanilla implementation, in simulations (with varying sample size) and real-data experiments. As it stands now, the paper fails to deliver the correct message.

---

> ### Author Response · Authors · 2025-08-23
> **Thank you for your feedback**
>
> We thank the reviewer for acknowledging our improvements in speed, flexibility, and the strengths of our theoretical and synthetic benchmark results. We respond to specific concerns inline:
>
> > Lack of direct comparison with original HyperDT on accuracy: The paper claims equivalence with HyperDT but does not report classification or regression accuracy comparisons with the original HyperDT on the same simulations.
>
> We have added the new Table 1 in our revised paper to address this concern.
>
> > No comparison to vanilla XGBoost/LightGBM: Since XGBoost and LightGBM are already highly optimized and accurate, it is unclear whether the hyperbolic wrapper actually adds value in accuracy or efficiency over these widely used baselines.
>
> The updated Table 2 (as well as 4 and 5 in the Appendix) use a new format where we compare (1) the base model, (2) the base model + preprocessing only, (3) full Fast-HyperDT. This demonstrates that, while using Euclidean XGBoost by itself is in fact more accurate than HyperDT, additional improvements come from pre- and postprocessing. The preprocessing improvement is large, whereas the postprocessing improvement is more modest, corroborating earlier results suggesting that exact midpoint placement is not a major driver of performance in HyperDT.
>
> > Tables 1 and 2 do not consider fair comparison: Accuracy improvements in these tables are primarily due to using boosted tree algorithms (e.g., XGBoost), not necessarily due to the hyperbolic method. Without vanilla XGBoost as a baseline, these comparison on accuracy are misleading.
>
> The updated Tables 2, 4, and 5 now include vanilla/base variants.
>
> > Theory is descriptive rather than rigorous: The "theorems" (Section 4) are mostly equivalence arguments and complexity analyses. They are not formal mathematical theorems.
>
> We have significantly updated Section 4 to be more rigorous. In particular we have completely rewritten our proof of Lemma 4.3.
>
> > Evaluation limited to synthetic data, no real-world datasets.
>
> We have added a new Section 5.4, which demonstrates very good performance on WordNet classification. We also compare to HoroRF and EPboost, two alternative hyperbolic random forest algorithms, on this task.
>
> > Add direct accuracy comparisons with HyperDT on the same datasets to confirm the equivalence claim.
>
> This is shown in the new Table 1.
>
> > Include runtime and accuracy benchmarks against vanilla XGBoost and LightGBM (without the hyperbolic wrapper) to show whether the wrapper introduces overhead or accuracy gains.
>
> This is shown in the new Tables 2, 4, and 5 (baselines have no pre/postprocessing).
>
> > Perform detailed simulation studies to isolate the impact of Klein projection and Einstein midpoint correction (e.g., comparing Scikit-Learn in Klein coordinates vs. Fast-HyperDT).
>
> This is shown in the new Tables 2, 4, and 5 (we benchmark preprocessing-only variants alongside full Fast-HyperDT and base models)
>
> > Expand experimental datasets to include real-world hyperbolic datasets for both speed and accuracy.
>
> In Section 5.4 we have benchmarked our method on WordNet classification.
>
> > Refine theoretical contributions by formalizing equivalence proofs and quantifying numerical approximation errors (e.g., due to midpoint adjustments).
>
> We have rewritten Section 4 substantially. While it is not clear how to quantify numerical approximation errors with regard to how they affect the splits learned by HyperDT and Fast-HyperDT, our benchmarks using a deterministic simplified baseline implementation of a decision tree (Figure 4) show that these come into play far less than our initial results on scikit-learn decision trees implied.
>
> > Clarify experimental tables: Clearly separate methods that are core contributions (Fast-HyperDT, Fast-HyperRF) from unrelated baselines (CO2, HHC, vanilla XGBoost).
>
> This is done in Tables 2, 4, and 5.
>
> > Overall, the most important part to me is that authors should convincingly show that their methods provide significant speedup, while not losing accuracy versus the vanilla implementation, in simulations (with varying sample size) and real-data experiments. As it stands now, the paper fails to deliver the correct message.
>
> We believe that, with the modifications we have put in, this message should now be clearer.

---

### Review · Reviewer_zt35 · 2025-07-31

**Summary Of Contributions:**

The paper presents a fast algorithm for Hyperbolic Decision Trees (HyperDT). HyperDT replaces axis-parallel hyperplanes in CART by hyperbolic hyperplanes, yet suffers from computational burdens. In this paper, the author leverages the Beltrami-Klein model to transform the data in hyperbolic spaces into open balls, where the original geodesics are transformed into straight lines. Therefore, classic decision tree methods can be applied directly to this transformed space and trained efficiently. Finally, the author proposed a method to adjust the midpoint to match the distance metric in the hyperbolic space. Experiments showed that the algorithm alleviates the training burden of HyperDT.

**Audience:**

Yes

**Audience Explanation:**

The paper provides a fast algorithm for hyperbolic DT, which is important for its practical application.

**Claims And Evidence:**

No

**Claims Explanation:**

1. The paper is poorly written, with a bunch of typos and an insufficient description of the experimental setting.
   * Page 3 footnote 1, why is it here?
   * Page 3 equations (4,6), please specify what $||\cdot||$ and $\langle\cdot,\cdot\rangle$ mean, e.g. Euclidean norm and inner product.
   * Page 3 sentence below equation (4): "related to through"
   * Page 3 sentence above equation (8):  should be "Klein disk at points **a** and **d**". Also please provide the reference.
   * Page 4 sentence below equation (14): shouldn't it be "are equidistant to the hyperplane $h_0cos(\theta_{naive})$"?
   * Page 8 equation (23), why **u,v**$\in \mathbb L^n_k$? Shouldn't it be $\mathbb B^n_k$.
   * Page 9 equation (26), should be $m_{\mathbb B}$ instead of $m_{\mathbb E}$.
   * Section 5: What computational resource do you use? How is the Gaussian mixture sampled? (e.g. what is the choice of the covariance?) How do you generate the response in the regression setting? How do you choose the curvature $K$?
2. The proof of Lemma 4.3 is not rigorous and needs to be clarified.
   * Please clarify equation (23), where do **u,v** actually belong to, and how the equation can be derived from equation (9), i.e. the definition of the Einstein midpoint.
   * Clarify why assuming **u,v** has zero coordinates in several positions.
   * Conclude the proof by deriving (21) based on your arguments.
3. The experimental setting is not detailed and needs to be extended.
   * The experiments on timing benchmarks need more different setups. For example, setting the maximum depth to 3 is too small. What is the dimension $d$, and how does it affect the speed? Did you include the preprocessing and the postprocessing time?
   * What is the SK-DT + Lorentz model? Is it a classic CART on the data $X\in \mathbb L^{n\times d}$? Please specify.
   * For CO2, HHC, LightGBM, and XGBoost, do you use fast HyperDT as the base model?
   * [1] seems to be relevant to your paper, and therefore should be discussed and compared. HoroRF should also be added to compare the performance.
   * Why do you use a fixed depth 3 even in section 5.3, rather than cross-validated $d$? Why don't you tune the hyperparameters?
   * Is there any pruning procedure required in HyperDT?
   * What is the error? Is it the training error or the test error? Are the hyperparameters cross-validated to produce more convincing results?
   * Error bars are not provided.
   * Only simulation datasets are considered, yet related works e.g. [1] and HoroRF, also consider real-world datasets such as WordNet. Adding real-world datasets would be helpful.
   * Lastly, the additional postprocessing procedure to adjust the middle point of the split threshold essentially has marginal effects on the results, which undermines the contribution of the paper.

[1] Suganthan, P. N., Kong, L., Snášel, V., Ojha, V., & Aly, H. A. H. Z. (2025). Euclidean and Poincare space ensemble Xgboost. Information Fusion, 115, 102746.

**Requested Changes:**

1. Fix all the typos and proofread the paper.
2. Clarify the proof of Lemma 4.3.
3. More detailed experimental settings and extensive experiments are required.

---

> ### Author Response · Authors · 2025-08-23
> **Thank you for your feedback**
>
> We thank the reviewer for their careful proofreading of our paper, and for their favorable assessment of our work's relevance to the TMLR readership. We address specific concerns inline:
>
> > The paper is poorly written, with a bunch of typos and an insufficient description of the experimental setting.
>
> All of these typos have been fixed, except for:
>
> > Page 8 equation (23), why u,v$\in \mathbb L^n_k$? Shouldn't it be $\mathbb B^n_k$.
>
> For this equation (now , $\mathbf{u, v}$ should in fact be elements of $\mathbb{L}^d_K$: we begin with two points in the Lorentz model, and the hyperbolic midpoint as given in hyperDT; we then show that this is equivalent to the fast-HyperDT midpoint as defined in the Klein model. Not in particular that we use $\delta_\mathbb{L}$, i.e. the hyperbolic distance. We have substantially rewritten Lemma 4.3 and its proof, and we hope it is clearer which space each point lives in. We have also adopted the convention of using $\mathbf{p, q}$ to refer to elements of $\mathbb{K}$, and continue using $\mathbf{u, v}$ to refer to elements of $\mathbb{L}$.
>
> > Section 5: What computational resource do you use? How is the Gaussian mixture sampled? (e.g. what is the choice of the covariance?) How do you generate the response in the regression setting? How do you choose the curvature $K$?
>
> We have added detailed descriptions of our benchmarking setup at the beginning of Section 5.1.
>
> > Please clarify equation (23), where do u,v actually belong to, and how the equation can be derived from equation (9), i.e. the definition of the Einstein midpoint.
>
> As discussed above, we have adopted variable naming conventions that differentiate between elements of $\mathbb{K}$ and $\mathbb{L}$ to address this ambiguity in the future.
>
> > Clarify why assuming u,v has zero coordinates in several positions.
>
> We have removed this assumption from the proof of 4.3, and added context about the sparsity of the normal vector in base HyperDT in Section 2.3.1 instead.
>
> > Conclude the proof by deriving (21) based on your arguments.
>
> Although the exact derivation of this identity is not tractable, our argument based on the (1) existence and (2) uniqueness of an equidistant point does show equivalence. We have rewritten the proof for Lemma 4.3 to be clearer on this point.
>
> > The experiments on timing benchmarks need more different setups. For example, setting the maximum depth to 3 is too small. What is the dimension $d$, and how does it affect the speed? Did you include the preprocessing and the postprocessing time?
>
> The extended Figure 2 contains additional experiments scaling depth and dimensionality. The description of the timing benchmarks in Section 5.2 now explicitly specifies that the timing includes pre- and post-processing time.
>
> > What is the SK-DT + Lorentz model? Is it a classic CART on the data $X\in \mathbb L^{n\times d}$? Please specify.
>
> That is correct. We have revised the way that results are presented in Table 2 to make it clearer that we consider 3 models: (1) base model, (2) base model + preprocessing to Klein coordinates, (3) base model + preprocessing + postprocessing midpoints (i.e. full Fast-HyperDT).
>
> > For CO2, HHC, LightGBM, and XGBoost, do you use fast HyperDT as the base model?
>
> Fast-HyperDT is a wrapper that can be placed around any decision tree-based model, provided the API for that model exposes the splits. We have carefully revised the use of the term "base model" in our paper to clarify this.
>
> > [1] seems to be relevant to your paper, and therefore should be discussed and compared. HoroRF should also be added to compare the performance.
>
> We have added a new Section 5.4 to address WordNet classification benchmarks. In this section, we compare to HoroRF and the EPBoost algorithm in the paper you have linked.
>
> > Why do you use a fixed depth 3 even in section 5.3, rather than cross-validated $d$? Why don't you tune the hyperparameters?
>
> The purpose of this paper is to demonstrate a method for integrating the core idea of HyperDT (geodesic splits) with other tree-based learning framework, not to achieve the absolute best performance on a set of benchmarks. While our method is scikit-learn compatible, and therefore supports hyperparameter tuning, comparing to baselines like HoroRF which are both computationally expensive and not scikit-learn standardized would require significant effort for hyperparameter tuning. We believe our results are sufficient to demonstrate that (1) using XGBoost can offer an improvement over scikit-learn, and (2) we are competitive with state-of-the-art methods on WordNet classification without further tuning.
>
> > Is there any pruning procedure required in HyperDT?
>
> No, but it is compatible with pruning in base models that happen to use it.

---

> > ### Author Response · Authors · 2025-08-23
> > **Thank you for your feedback (continued)**
> >
> > > What is the error? Is it the training error or the test error? Are the hyperparameters cross-validated to produce more convincing results?
> >
> > The error is test error, except where otherwise noted (i.e. in Figures 3 and 4, which consider predictions on the training and test sets separately). We use base hyperparameters, per our justification above.
> >
> > >  Error bars are not provided.
> >
> > We have added error bars and confidence intervals to all of our plots and tables.
> >
> > > Only simulation datasets are considered, yet related works e.g. [1] and HoroRF, also consider real-world datasets such as WordNet. Adding real-world datasets would be helpful.
> >
> > We have added Section 5.4 to address WordNet classification specifically. We find that our method is competitive with state-of-the-art methods in this setting.
> >
> > > Lastly, the additional postprocessing procedure to adjust the middle point of the split threshold essentially has marginal effects on the results, which undermines the contribution of the paper.
> >
> > We have more thoroughly characterized the effect of pre- and post-processing in our revised Table 2, and added a discussion to the conclusion suggesting to practitioners that they can gain many of the benefits of Fast-HyperDT by simply working with decision trees in the Klein model. Nonetheless, the best-performing methods do rely on postprocessing as well as preprocessing.

---

### Review · Reviewer_PRs9 · 2025-08-02

**Summary Of Contributions:**

This paper proposes a fast implementation of HyperDT, a decision tree in hyperbolic space. The key idea is to map the input data using the Beltrami–Klein model, which enables the use of standard Euclidean decision tree algorithms in the projected space. Theoretical analysis proves the equivalence between this approach and the original HyperDT. Thanks to the well-developed decision tree libraries with optimized implementations such as Scikit-learn, the proposed method achieves a speedup of more than three orders of magnitude compared to the original HyperDT algorithm on datasets with around 30,000 samples.

**Audience:**

Yes

**Audience Explanation:**

Since tree-based learners such as CART and Random Forest are widely used, and their hyperbolic counterpart, HyperDT, also holds practical value, I believe the proposed approach, offering a faster implementation, is of interest to the ML community.

**Broader Impact Concerns:**

I do not have any concerns.

**Claims And Evidence:**

No

**Claims Explanation:**

**Strengths**
1. The main contribution of this paper, a simple yet effective trick for accelerating HyperDT, is useful and has the potential to be widely adopted in various applications.
2. The paper is clearly written and well-organized. It easy to follow.
3. It is nice that the equivalence between the proposed method and the original HyperDT has been carefully analyzed and theoretically proven. This result can serve as a foundation for further development of tree-based learners in hyperbolic spaces.

**Weaknesses**
1. The reason for the substantial speedup reported should be analyzed in more detail. Since the theoretical time complexity of the proposed method and the original HyperDT is the same, their runtime should also be comparable in theory. If the speedup is simply due to implementation differences, such as the optimization level of Scikit-learn versus that of HyperDT, then it suggests that HyperDT itself could also be significantly accelerated through better coding practices. This would diminish the impact of the current contribution. A more fine-grained analysis is therefore needed. For instance:
	- By examining the source code of Scikit-learn’s RandomForestClassifier, which specific components contribute to the performance gap?
	- How does the performance compare when using a simple, non-optimized implementation of a standard decision tree instead of Scikit-learn? An empirical comparison with such a baseline would add value.
2. As the authors acknowledge, there are experimental discrepancies between HyperDT and Fast-HyperDT. While this is speculated to be due to tiebreaking rules or numerical precision issues, and some synthetic data experiments are provided, the issue remains unresolved. This gap is critical because the user may be unsure whether Fast-HyperDT will produce equivalent results to the original, and there is a possibility that Fast-HyperDT could perform worse on certain datasets. Further analysis is necessary, and ideally, a method should be proposed to ensure Fast-HyperDT produces identical results to HyperDT.
3. Minor typographical issues:
	- In Abstract, "HyperDTas" -> "HyperDT as"
	- Please remove the footnote on P.3.

**Requested Changes:**

Please address the three concerns outlined in the Weaknesses section.

---

> ### Author Response · Authors · 2025-08-23
> **Thank you for your feedback**
>
> We thank reviewer **PRs9** for their detailed feedback, and for their favorable assessment of our contributions, quality of writing, analysis, and relevance to the TMLR readership. We address the reviewer's concerns inline:
>
> > The reason for the substantial speedup reported should be analyzed in more detail. Since the theoretical time complexity of the proposed method and the original HyperDT is the same, their runtime should also be comparable in theory. If the speedup is simply due to implementation differences, such as the optimization level of Scikit-learn versus that of HyperDT, then it suggests that HyperDT itself could also be significantly accelerated through better coding practices. This would diminish the impact of the current contribution. A more fine-grained analysis is therefore needed. For instance:
> > * By examining the source code of Scikit-learn’s RandomForestClassifier, which specific components contribute to the performance gap?
>
> Scikit-Learn's random forest and decision tree classes are written in Cython, which makes a detailed analysis of this question challenging and time-consuming. We have added language to our paper (at the beginning of Section 5.2) explicitly specifying that our timing benchmarks include preprocessing + training + postprocessing for this reason, and have also expanded the timing benchmarks in Figure 2 to include scaling over maximum depth and dimensionality. We know that the asymptotic behavior of Fast-HyperDT is equivalent to a Euclidean decision tree, so these runtime optimizations must be from constant-time improvements in the underlying implementation.
>
> > * How does the performance compare when using a simple, non-optimized implementation of a standard decision tree instead of Scikit-learn? An empirical comparison with such a baseline would add value.
>
> We have added a comparison to a non-optimized baseline in Figure 4. The purpose of this comparison is to explore the difference in splitting behavior between HyperDT and the Fast-HyperDT variants built on the Scikit-Learn DecisionTreeClassifier class. We find that the Fast-HyperDT does not represent a significant speedup over HyperDT in this case, which supports the idea that Scikit-Learn implementation details drive the speedup we observe.
>
> > As the authors acknowledge, there are experimental discrepancies between HyperDT and Fast-HyperDT. While this is speculated to be due to tiebreaking rules or numerical precision issues, and some synthetic data experiments are provided, the issue remains unresolved. This gap is critical because the user may be unsure whether Fast-HyperDT will produce equivalent results to the original, and there is a possibility that Fast-HyperDT could perform worse on certain datasets. Further analysis is necessary, and ideally, a method should be proposed to ensure Fast-HyperDT produces identical results to HyperDT.
>
> As we mention above, Figure 4 now includes a comparison with a more "apples-to-apples," non-optimized baseline. This comparison shows very high agreement both in predictions and exact splits (100% for maximum depths <= 9, and 99.4%+ for unbounded maximum depths), except near the boundaries of the Klein disk where HyperDT incurs some numerical instability in its midpoint computation.
>
> > In Abstract, "HyperDTas" -> "HyperDT as"
>
> We have fixed this.
>
> > Please remove the footnote on P.3.
>
> We have fixed this.

---

### Decision · Action_Editor_HfAw · 2025-10-04

**Recommendation:** Reject

**Additional Comments:**

The paper has potential to contribute in a significant way to the development of efficient decision trees in hyperbolic space, however it is the opinion of two reviewers out of three that the paper needs another round of revision because of major modifications in all areas (theory, experiments, presentation). Especially presentation needs to be improved, by providing more experimental evidence involving real data and a better and more articulated justification of the advantages of the proposed approach. Authors are warmly encouraged to improve presentation following the above lines, and to resubmit the updated version of the paper for a new round of review.

**Audience:**

Yes

**Audience Explanation:**

Decision trees are one of the workhorses of machine learning, so any paper that contributes to the development of them is of interest for a large audience.

**Claims And Evidence:**

No

**Claims Explanation:**

Although the aim of the paper seems to be compelling, i.e.  to provide a much faster implementation of HyperDT, a decision tree in hyperbolic space, the current version of the paper substantially changed the original submission while not completely elucidating all the advantages of the proposed implementation.

**Resubmission Of Major Revision:**

The authors may consider submitting a major revision at a later time.